# Mechanical Stretch Induced Skin Regeneration: Molecular and Cellular Mechanism in Skin Soft Tissue Expansion

**DOI:** 10.3390/ijms23179622

**Published:** 2022-08-25

**Authors:** Yaotao Guo, Yajuan Song, Shaoheng Xiong, Tong Wang, Wei Liu, Zhou Yu, Xianjie Ma

**Affiliations:** 1Department of Plastic Surgery, Xijing Hospital, Fourth Military Medical University, Xi’an 710032, China; 2Department of The Cadet Team 6, School of Basic Medicine, Fourth Military Medical University, Xi’an 710032, China

**Keywords:** skin soft tissue expansion, mechanical stretch, signaling pathway, cell adhesion molecules, transcriptome sequencing, skin stem cell, mesenchymal stem cells

## Abstract

Skin soft tissue expansion is one of the most basic and commonly used techniques in plastic surgery to obtain excess skin for a variety of medical uses. However, skin soft tissue expansion is faced with many problems, such as long treatment process, poor skin quality, high retraction rate, and complications. Therefore, a deeper understanding of the mechanisms of skin soft tissue expansion is needed. The key to skin soft tissue expansion lies in the mechanical stretch applied to the skin by an inflatable expander. Mechanical stimulation activates multiple signaling pathways through cellular adhesion molecules and regulates gene expression profiles in cells. Meanwhile, various types of cells contribute to skin expansion, including keratinocytes, dermal fibroblasts, and mesenchymal stem cells, which are also regulated by mechanical stretch. This article reviews the molecular and cellular mechanisms of skin regeneration induced by mechanical stretch during skin soft tissue expansion.

## 1. Introduction

Skin soft tissue expansion is one of the most essential and common techniques in plastic surgery, which can provide large amounts of extra skin tissue with similar color, texture, and thickness for a variety of uses [1]. During this procedure, an inflatable silicone expander is implanted under the skin. With regular injections of saline, the progressively enlarging expander applies tension to the skin and continuously promotes cell and skin growth. Mechanical stretch stimulation is the dominant factor to induce expanded skin biological growth during tissue expansion. At present, skin soft tissue expansion is widely used for both adults and children [2,3] and applied in various conditions, such as breast reconstruction [4], ear reconstruction [4], burn deformities [4], bone graft [5], removal of giant congenital melanocytic nevi [6], and other medical applications.

Although skin soft tissue expansion is widely used, due to the slow growth of the expanded skin, the procedure can go on for months. At the same time, there are problems, such as poor skin quality, high retraction rate, and complications, caused by expansion. The mechanistic study of mechanical-stretch-induced skin regeneration during skin soft tissue expansion is the foundation to solve these problems. Therefore, researchers have conducted in-depth studies on the complex mechanism of skin soft tissue expansion and obtained many important findings. Skin soft tissue expansion involves a complex mechanobiology process, including dynamic fluctuations in force and shape that happen in skin, which is similar to tissue engineering [7]. The major biological responses to mechanical stretch exerted by skin expansion, including biological growth, elastic stretching, displacement, and mechanical creep, are based on the enlargement of tissue expander. Biological growth is the most important and long-lasting biological response and produces the majority of the newly grown skin, which is attributed to the complex regulation of mechanical stimulation at the molecular and cellular levels. At the molecular level, the main result of mechanical stimulation is the activation of multiple signaling pathways and cellular adhesion molecules, especially β1 integrin and E-cadherin, play an important role during this process. In addition, transcriptome sequencing results reveal a large number of differentially expressed genes (DEGs) induced by mechanical stretch, which finally leads to changes in cell behaviors and states, including cell proliferation, differentiation, migration of keratinocytes, fibroblasts, and mesenchymal stem cells.

During skin soft tissue expansion, changes occur in the epidermis, dermis, adipose tissue, muscles, blood vessels, and skin accessory structures at the site of expansion. In the epidermis, the thickness and cell density increase and the basal keratinocytes are active in mitosis [8]. In the dermis, the thickness becomes thinner, the collagen density increases, and the collagen fibers are stretched, mostly parallel to the surface of the expander. Part of the collagen fibers is broken and arranged in disorder and the number of active fibroblasts also increases [9]. Transmission electron microscopy shows that the nuclear membranes of the cells in the epidermis and dermis are folded and the number of organelles increases, indicating active proliferation [8,9]. The thickness of subcutaneous fat decreases and a large number of mitochondria of different shapes and sizes is observed at the periphery of muscle fibers [8]. The muscles after expansion are thinner. A fibrous capsule mainly composed of collagens and fibroblasts is located under the muscle and enclosed around the tissue expander [8]. The density of blood vessels in the expanded skin increases [10]. The hair follicles of the expanded skin are active and the hair growth accelerates significantly [11].

This article reviews the molecular mechanisms (signaling pathways, cell adhesion molecules, and transcriptome change) and cellular changes in keratinocytes, fibroblasts, and mesenchymal stem cells during skin soft tissue expansion.

## 2. Clinical Problem

In clinical practice, the main problems in skin soft tissue expansion are long operation cycle, low efficiency of skin expansion, and poor skin quality. The slow growth of the skin results in a prolonged process that usually takes two to six months after the implantation of the expander before the second stage of surgery. Low efficient skin expansion leads to too-small areas of extra skin to meet the requirement for tissue defect repair. Therefore, clinically, it is expected that more additional skin can be obtained with less expansion time, which means accelerated skin regeneration during tissue expansion. To achieve this goal, at the molecular level, genes involved in cell division, differentiation, metabolism, and angiogenesis should be up-regulated and signaling pathways that promote tissue regeneration should be activated. At the cellular level, the proliferation and synthesis ability of various cells should be improved, which is conducive to the increase in cell number and the accumulation of extracellular matrix in the dermis. At the same time, stem cells should be motivated or recruited to differentiate into various types of skin cells to promote skin regeneration. All the molecular and cellular changes finally facilitate expanded skin regeneration, thus, obtaining more regenerated skin with similar appearances and histological characteristics to the original skin.

## 3. Molecular Mechanism

During skin soft tissue expansion, a complex molecular regulation mechanism takes place in skin tissue under mechanical stretch. Multiple types of signaling, including YAP/TAZ signaling, MAPK-ERK signaling, Wnt/β-catenin signaling, and AP-1, have been found to be activated. Cellular adhesion molecules, which are capable of sensing mechanical stimuli, have been found to be common upstream regulatory molecules in multiple signaling pathways and may be the initiation factors of numerous intracellular reactions. At the same time, more and more transcriptome sequencing data reveal the gene expression changes during skin soft tissue expansion.

### 3.1. Intracellular Signaling under Mechanical Stretch

#### 3.1.1. YAP/TAZ Signaling

It is well established that Yes-associated protein (YAP) and transcriptional coactivator Tafazzin (TAZ) are two core cotranscription factors in the hippo pathway, which function in cell growth and fate [12,13]. When activated, YAP/TAZ migrates to the nucleus and bind to TEAD transcription factor family, which regulate gene expression to promote tissue growth and inhibit apoptosis [14]. YAP/TAZ was first found to be regulated by cell–cell contact and cell polarity and then was recognized as important proteins in intracellular mechanical signaling [15]. Aragona et al. found that mechanical cues were one of the dominant factors affecting YAP/TAZ activity and that a mechanically stressed cytoskeleton is essential for the introduction of multiple signals (including Wnt and GPCR) into YAP/TAZ [16]. Dupont et al. studied the role of YAP/TAZ in the mechanotransduction in mammary epithelial cells (MEC) and human lung microvascular endothelial cells (HMVEC) [17]. They found that YAP/TAZ activity is regulated by extracellular matrix stiffness and cell geometry. YAP/TAZ is activated in cells grown on the high-stiffness extracellular matrix, but inhibited in the low-stiffness extracellular matrix and this regulation is mediated by cytoskeletal tension. Das et al. found that the regulation of cytoskeleton integrity in mouse embryonic fibroblasts on YAP activity was superior to that on actomyosin contractility [18]. In vitro, inhibition of actin cytoskeleton suppresses YAP/TAZ transcriptional activity, whereas induction of F-actin polymerization and stress fiber formation by activation of diaphanous protein (DIAPH1) facilitates YAP/TAZ activity [17].

During the process of skin soft tissue expansion, mechanical stretch promotes the activation of YAP/TAZ and, thus, results in skin regeneration [17,19]. Wang et al. revealed that short-term mechanical stretch induces the activation of YAP in interfollicular epidermal stem cells and suppression of YAP is observed under long-term mechanical stretch, implying that YAP/TAZ is subject to complex regulation during skin expansion [13]. YAP/TAZ is regulated by a variety of signaling types, including hippo, integrins, and mechanical stimulation [20]. Though the hippo pathway suppresses the activation of YAP/TAZ through MST1/2 and LATS1/2, restricting their translocation to the nucleus, the mechanical stimulation promotes nuclear translocation of YAP/TAZ. After receiving the mechanical signals, integrin-recruited FAK and Src either activate YAP/TAZ directly or suppress the activity of LATS1/2 to promote the translocation of YAP/TAZ to the nucleus (Figure 1) [21]. Recently, Xue et al. studied the function of YAP in skin soft tissue expansion and found that high levels of YAP nuclear translocation contribute to more active keratinocyte proliferation, thicker epidermis, and faster skin growth in Krt5-rtTA; tetO-YAPS112A mice and that deletion of YAP results in reduced epidermal thickness and slow skin growth in Lgr6Cre; YAPflox mice, in which the expression of TAZ does not change [22].

#### 3.1.2. MAPK-ERK Signaling

The generic mitogen-activated protein kinases (MAPK) pathway works mainly through a three-stage protein kinase cascade reaction: MAP2K kinase (MKKK/MAP3K), MAPK kinase (MKK/MAP2K), and MAPK. It consists of four branches, named by their MAPK components: the extracellular-signal-related kinases (ERK1/2) signaling, Jun aminoterminal kinases (JNK1/2/3) signaling, p38 signaling, and ERK5 signaling. The activation of ERK signaling is able to facilitate proliferation and migration of fibroblasts and keratinocytes, vascularization, and skin regeneration [23,24,25]. Meanwhile, ERK1/2 signaling can be significantly activated in human dermal fibroblasts by high-frequency repetitive stretch, implying its essential role in skin soft tissue expansion [26]. Recently, Qiang et al. found that ERK signaling also promotes proliferation and differentiation of keratinocytes and activation of fibroblasts by mediating TNF-induced autophagy [27]. However, abnormal activation of the MAPK pathway breaks epidermal barrier integrity and leads to pathological changes in the skin by upregulating the expression of proinflammatory factors [28,29,30].

The ERK signaling responds to mechanical stretch and a variety of growth factors, including FGF, EGF, and VEGF, which is mainly mediated by Ras-Raf-MEK-ERK cascade [31,32,33]. Through in vitro cell cyclic stretching experiments (1 Hz, 120% in length), Wang et al. found that mechanical stretch activated ERK cascade by inducing focal adhesion kinase (FAK) tyrosine phosphorylation (Figure 1) [34]. Phosphorylated FAK subsequently leads to the activation of GRB2, SOS. Then the activated SOS turns the non-activated GDP-bound Ras into the activated GTP-bound Ras, which initiates Raf-MEK-ERK cascade subsequently. Activated ERK1/2 translocates into the nucleus and activates AP-1, which regulates the transcription of immediate early genes [35]. ERK regulates AP-1 activity through c-Fos during transcription and post transcription. On the one hand, ERK enhanced the activity of ternary complex factor (TCF), which combined with serum response element (SRE) on c-Fos gene to promote c-Fos transcription [36]. On the other hand, ERK and its downstream MAPKAP kinase RSK phosphorylate c-Fos to make it more stable to bind c-Jun [37]. CCN2, also known as connective tissue growth factor, has been shown to be an activator of the FAK-ERK pathway, enhancing keratinocyte migration [38].

#### 3.1.3. Wnt/β-Catenin Signaling

The Wnt pathway is a highly conserved signaling pathway that plays an important role in early embryonic development, organogenesis, tissue regeneration, and other physiological processes. In the skin, it is well known that Wnt signaling is involved in epidermal differentiation and stratification during embryonic development, hair follicle development, maintenance of normal epidermal spinous layer, and skin homeostasis [39,40,41]. Basal keratinocyte cell proliferation in interfollicular epidermal (IFE) and hair follicle development is strictly controlled by Wnt/β-catenin signaling. Choi et al. found that β-catenin deletion or ectopic expression of Dkk1, a Wnt/β-catenin inhibitor, will cause rapid hair follicle regression and loss of hair follicle stem cells [42]. Lim et al. found that inhibition of Wnt/β-catenin signaling in keratinocytes suppressed basal keratinocytes proliferation and premature differentiation [43].

A number of studies also show that activating Wnt/β-catenin signaling by mechanical stretch contributes to expanded skin regeneration. In bone marrow mesenchymal stem cells, cyclic strain induces osteogenic differentiation through Wnt/β-catenin [44]. In hepatocellular carcinoma, β-catenin can sense the stiffness in the extracellular matrix [45]. In in vitro cell experiments, Samuel et al. found that higher stiffness in the extracellular matrix leads to more mechanical loading, resulting in intracellular activation of Rho/ROCK, which promoted the proliferation of epidermal cells through β-catenin-dependent and actomyosin contractility dependent ways [46]. In vivo, Wnt/β-catenin signaling also promotes expanded skin growth by regulating the differentiation of hair follicle stem cells (HFSCs). Our previous studies found that after being injected into the expanded skin, HFSCs can differentiate into endothelial cells, epidermal cells, and the outer root sheath cells of hair follicle, indicating that mechanical stretch can induce HFSC differentiation to facilitate expanded skin regeneration [47]. Our study and others also found that the Wnt pathway is activated in expanded skin, suggesting that mechanical stretch promotes skin regeneration through the Wnt pathway [48,49]. In addition, Ledwon et al. recently found that mechanical stretch induces Wnt signaling activation and accumulation of β-catenin in basal keratinocytes, leading to cell proliferation and epidermal growth [50]. In addition, langerhans cells regulate expanded skin regeneration through the Wnt pathway, too [49].

Mechanical stretch can induce Wnt activity through Wnt-protein-dependent and -independent ways (Figure 2). The Wnt-protein-dependent way indicates that, mechanically, it can up-regulate the expression of Wnt protein directly to activate the Wnt pathway [48]. The Wnt-protein-independent way is associated with degradation of E-cadherin induced by mechanical stretch. There are two pools of β-catenin in cells, the E-cadherin binding pool and the cytoplasmic pool. Normally, β-catenin in these two pools functions independently, but under certain conditions, such as mechanical stretch, degradation, or down-regulation of E-cadherin, the bound β-catenin is released into the cytoplasm, which activates the Wnt pathway [51]. The accumulated β-catenin in the nucleus eventually binds to TCF/LEF (T-cell factor/lymphoid enhancer factor) and regulates Wnt target genes through them [52]. In summary, loss of E-cadherin and activation of Wnt/β-catenin signaling promote epithelial-mesenchymal transitions (EMT) [51]. This is consistent with the results observed in skin soft tissue expansion [53].

#### 3.1.4. AP-1

Activator Protein 1 (AP-1), a transcriptional regulator, is composed of Fos and Jun family members of DNA-binding proteins. It converts a variety of extracellular signals, such as growth factors, neurotransmitters, polypeptide hormones, and physical and chemical stresses, through evolutionary conserved signaling pathways, such as MAPK, Wnt, and TGF-β [54]. In the skin, AP-1 is involved in regulating keratinocyte and fibroblast proliferation [55]. It plays a role downstream of the MAPK-ERK pathway and promotes keratinocyte and dermal fibroblast proliferation [27]. AP-1 has also been implicated in wound healing and re-epithelialization [56]. Generally, Jun is considered to be a positive regulator, while JunB and JunD are considered negative regulators [20]. Angel et al. suggested that c-Jun, JunD, and Fra-1 might function in keratinocyte proliferation and the early stage of differentiation [55]. AP-1 is also an important mechanosensitive protein and has been found to be up-regulated in a variety of tissues and cells in response to mechanical stretch, such as bladder muscle cells, osteoblasts, lung parenchyma, amnion cells, and vascular smooth muscle cells [20]. Papadopoulou et al. found that both cyclic and static mechanical strains could up-regulate AP-1 members (c-Fos, c-Jun) in vitro in cultured human periodontal ligament fibroblasts [57]. This up-regulation has also been observed during skin soft tissue expansion. Aragona et al. found that transcription of AP-1 family members (Fos, FosB, JunB, FosL1, also known as Fra-1) was up-regulated in basal keratinocytes in expanded skin [58]. Assay for targeting accessible chromatin with high-throughput sequencing (ATAC-SEQ) results showed that the AP-1 chromatin region was unbound, also indicating that its transcription increased under the mechanical-stretch condition [58]. The target genes of AP-1 regulate the cell cycle into S phase and control cell proliferation. Through ChIP-seq, Zanconato et al. found that YAP/TAZ/TEAD and AP-1 form transcription factor complexes, which combine with complex regulatory elements to jointly regulate cell-proliferation program in basal keratinocytes [59].

### 3.2. Cellular Adhesion Molecules That Sense and Transmit Mechanical Signals

Mechanical conduction is the most important process during skin soft tissue expansion. As the volume of the implanted expander increases, the mechanical stretch applied to the local skin is transmitted to each cell via cell–cell and cell–extracellular matrix adhesion. Cell adhesion molecules, located on the surface of cell membrane, are mechanosensitive proteins. Cellular adhesion molecules transduce mechanical signals by changing quantity, conformation, and clustering. Clustering occurs in both integrins and cadherins, which can extend adhesion life and facilitate resistance to high-mechanical forces [60,61]. Cell adhesion molecules also connect to the cytoskeleton and actin affects the adhesion strength, suggesting that the cytoskeleton may be involved in intracellular mechanical signal transduction [62,63]. During tissue expansion, integrin and cadherin play an especially important role in conducting mechanical signals.

β1 integrin has been shown to mediate adhesion, regulate cell migration and re-epithelialization, and participate in mechanical transduction in skin [64]. Several studies have demonstrated that mechanical stretch can directly activate β1 integrins independently of ligands, thus, affecting cell division [65,66]. Integrin clustering can form larger focal adhesions and recruit the signaling molecule focal adhesion kinase (FAK), which, in turn, recruits and activates Src and GRB2, and finally, GRB2 mediates the activation of downstream signals of MAPK (Figure 1) [64,67]. β1 integrin was also found to be highly correlated with YAP/TAZ signaling [68]. FAK and Src activate YAP to promote cell proliferation in various ways (Figure 1) [21]. In addition, Jolanta et al. found that β1 integrins also regulate cell division by repositioning in response to mechanical stretch [69]. The amount of β1 integrin on the surface of mitotic precursors increased significantly one hour after tissue expansion and returned to normal 24 h later as local skin adapted to mechanical stimulation. In addition to mechanical transduction, other integrin family members can promote keratinocyte proliferation and adhesion by binding to matricellular protein, such as CCN1 and CCN2 [38,70,71].

E-cadherin, a calcium-dependent cell adhesion molecule, consists of five tandemly repeated domains in the extracellular domain, a transmembrane domain, and an intracellular domain. In general, E-cadherin loss is considered a marker of epithelial mesenchymal transition (EMT) in cancer [72]. E-cadherin links the actin cytoskeleton via α-catenin and β-catenin and interacts with the Wnt signaling pathway [51]. Huang et al. recently found that E-cadherin is down-regulated during tissue expansion and regulates EMT through the β-catenin-FOXO1-KLF4 pathway [53]. However, Lewis et al. believed that down-regulation of E-cadherin during mechanical stretching was detrimental to epidermal integrity [73]. Desmoglein-3, another member of the cadherin family, was also found to be involved in regulating the response of keratinocytes to mechanical forces [74].

### 3.3. Ion Channels

Ion channels play an important role in mechanical transduction, which can transform extracellular mechanical signals into intracellular chemical signals. Although ion channels have not been studied in soft tissue expansion so far, their role in skin trauma has been extensively studied and has shown significant effects, so we can expect them to function in skin soft tissue expansion. Various ion channels have been found to be involved in mechanotransduction processes, such as PIEZO family, transient receptor potential (TRP) channels, OSCA/TMEM63 channels, and two-pore potassium channel (K2P) family [75]. They are involved in many important physiological processes in the body, such as regulating cellular responses to mechanical stimuli. He et al. found mechanical stretch promotes hypertrophic scar formation through Piezo1 [76]. Various kinds of transient receptor potential channels of the vanilloid subtype (TRPV) are mechanosensitive and are implicated in distinct physiological and pathological processes [77]. Meanwhile, the activation of TRPV3 can regulate inflammation and enhance the proliferation of skin keratinocyte [78,79]. In addition, Wei et al. found that the activation of TRPA1 promotes skin regeneration in adult mammalians [80].

### 3.4. Transcriptome Changes Induced by Mechanical Stretch

Transcriptome sequencing technology based on next-generation sequencing makes it possible to obtain the transcriptome information of specific tissues or cells comprehensively and quickly. In recent years, transcriptome sequencing technology has been increasingly used in the study of skin soft tissue expansion to explore the gene expression changes at the RNA level and reveal its molecular mechanism. Studies by our group and others have identified a number of hub genes differentially expressed in expanded skin, mainly affecting inflammatory response, tissue remodeling, and cytoskeleton and cell contraction [48,81,82]. Mechanical stimulation is the main cause of these DEGs.

Numerous studies have demonstrated that inflammatory response is essential for maintaining skin homeostasis and promoting skin regeneration [83,84]. Clearance of macrophages inhibits skin expansion [85]. In addition, mechanical stress and immunological response regulate skin regeneration through complex interaction, which is mimicked by skin tissue engineering [86]. It was found that a number of inflammation-related genes are up-regulated in expanded skin tissues, especially in the early stages of expansion, such as CXCL1, CXCL2, CXCL8, CXCR2, CCL20, CCR6, C3, C4A, and C5AR1 [48,81]. Chemokines and their receptors mainly play an important role in inducing leukocyte migration and are important proteins in regulating immune response. Keratinocytes express CXCL1, CXCL2, CXCL8, and CCL20, which recruit neutrophils and are up-regulated in psoriasis to increase inflammation [87]. CXCL1, CXCL2, and CXCL8 are mechanoresponsive proteins that are up-regulated by mechanical stimulation in keratinocytes and fibroblasts [88,89,90]. CXCR2, a G-protein-coupled receptor, is a common receptor for CXCL1, CXCL2, and CXCL8, which can promote keratinocyte proliferation and angiogenesis [91,92]. CXCR2 not only activates MAPK-ERK signaling and β1 integrin-FAK, but also is involved in the hypoxia-related HIF-1 signaling pathway [92]. The binding of CXCR2 to HIF-1 can enhance the tolerance of cells to anoxic environment [93]. KEGG pathway analysis also found that the HIF-1 signaling pathway is enriched in tissue expansion [81]; thus, the communication between CXCR2 and HIF-1 signaling may play a potential vital role in cell adaptation to the hypoxic environment after expansion. The CCL20-CCR6 axis functions in regulating the cutaneous immune response by driving the migration of different kinds of inflammatory cells, including B cells, immature dendritic cells, innate lymphocytes (ILC), regulatory CD4 T cells, and Th17 cells, and activation of the CCL20-CCR6 axis has been observed in psoriasis [94]. In skin soft tissue expansion, the mechanism of CCL20-CCR6 axis promoting skin regeneration may be its participation in inducing polarization of M2 macrophages [48]. Keratinocytes express various complement components and receptors, such as C3, C4, CR1, and C5AR1, and the activation of the complement system is involved in the occurrence of various inflammatory skin diseases [95]. Recent studies also found that C5/C5AR1 increased the sensitivity of the paw to mechanical stimulation in mice through TRPV1 [96].

After the expander is enlarged, the tissue remodeling process begins at an early stage (1 h) with the up-regulation of tissue-remodeling-related genes (MMP1, 2, 9, TIMP1), as the previously stable skin structure is destroyed by mechanical stretch [81]. Matrix metalloproteinases (MMP) are a zinc-dependent endopeptidase family involved in the degradation of the extracellular matrix and the regulation of inflammatory processes [97]. In the skin, MMPs repair fibrosis by degrading collagens, allowing scar tissue to dissolve [98]. A large number of studies has proved that MMPs (including MMP1, 2, 9) can also promote angiogenesis [99,100,101]. In the bladder, increased hydrostatic pressure will result in mechanical stretch stimulation similar to skin soft tissue expansion. He et al. found that mechanical stimulation could promote the expression of MMPs in bladder tissue, indicating that the expression of MMPs was induced by mechanical stretch [102]. TIMP1 is a natural inhibitor of most MMPs. The MMP/TIMP ratio is thought to determine the degree of ECM protein degradation and tissue remodeling [103]. Although both TIMP1 and MMPs were up-regulated during skin soft tissue expansion, MMPs were up-regulated more significantly [81]. A similar situation was found in periodontal ligaments during orthodontics [104]. Therefore, we believe that mechanical stretch may promote the proportion increase in TIMP1/MMPs in skin soft tissue expansion to regulate the tissue-remodeling process.

Another piece of evidence that mechanical stretch regulates gene expression and induces skin growth is the collective down-regulation of genes associated with cytoskeleton and cell contraction, such as troponins (TNNT1, TNNT3, TNNC2, TNNI2), myosins (MYH7), tropomyosins (TPM1, TPM2), actinins (ACTN2, ACTN3), actins (ACTA1), myozenin (MYOZ1), nebulin (NEB), and FLNC [48,81]. Cytoskeleton is the main structure of mechanical stress transduction in cells. The cytoskeleton can respond to mechanical stress by changing its shape and participate in cell contraction, migration, division, adhesion, and other processes [105]. Epithelial cells can adapt to mechanical stress by changing the shape of the cytoskeleton [106]. The down-regulation of these cytoskeleton proteins affects the structure, cross-linking, and motility of the cytoskeleton, leading to decreased cell contraction ability. Currently, the role of mechanical-stretch-induced down-regulation of cytoskeleton protein in skin soft tissue expansion is still unknown.

Several transcription factors (Fos, FosL1, LEF1, TCF7, HMGA1, ELF5, IRF4, SPI1) have also been identified that may be crucial in skin soft tissue expansion [48,107]. FOS and FOSL1 are members of the Fos gene family and the proteins encoded by them can dimerize with Jun family proteins to form the transcription factor complex AP-1. AP-1 engages in inflammatory and mechanical transduction processes and regulating cell proliferation and differentiation in the skin [20]. In addition, up-regulation of Fos induces down-regulation of E-cadherin, which is consistent with previous findings [108]. LEF1, TCF7, and HMGA1 are all involved in the Wnt/β-catenin signaling pathway and can regulate cell proliferation through the Wnt/β-catenin signaling pathway [109,110,111,112]. We discussed previously that mechanical stretch can induce the activation of Wnt/β-catenin signaling pathway to promote skin soft tissue expansion. ELF5, IRF4, and SPI1 are important transcription factors that regulate the differentiation, activation, and proliferation of immune cells and are believed to be involved in the foreign body response to silicone during tissue expansion [107].

In additional, Liu et al. studied the role of circRNA in expanded skin suffering from mechanical stretch [113]. In total, 48 circRNAs were found differentially expressed in expanded and unexpanded skin, involving multiple signaling pathways regulating hair follicle stem cells, such as the expression of the adenosine triphosphate binding cassette (ABC) transporter, suggesting the important role of hair follicle stem cells in skin expansion. These studies collectively analyzed the average transcription of genes in all cells in the expanded skin, but they provide great help in revealing the mechanisms of mechanical stretch promoting skin regeneration during skin soft tissue expansion. Aragona et al. conducted single-cell RNA sequencing of basal cells using a multidisciplinary approach that combined clonal analysis with quantitative modeling and single-cell RNA sequencing, in which they found that different types of cells responded differently to mechanical stretching [58].

### 3.5. Molecular Strategies to Improve Skin Soft Tissue Expansion

During skin soft tissue expansion, mechanical stimulation leads to an extremely complex molecular response. Fortunately, some of these critical molecular responses have been identified. Numerous kinds of activators or inhibitors corresponding to previously described signaling pathways, such as YAP/TAZ signaling, MAPK-ERK signaling, and Wnt/β-catenin signaling, have been developed. Regulating these pathways may improve expanded skin regeneration. Although the role of these significant signaling pathways is important, the potential role of other signaling pathways should not be ignored; further, the crosstalking of different signaling pathways during skin soft tissue expansion deserves further study. The function of various ion channels is another important research direction. There is no doubt that ion channels are important mediators of mechanotransduction. In-depth elucidation of the mechanism of ion channel action in skin and soft tissue expansion may provide a novel therapeutic strategy to regulate ion channel activity.

## 4. Cellular Mechanism

### 4.1. Keratinocyte in Epidermis under Mechanical Stretch

Keratinocytes in the basal layer divide and begin to differentiate and migrate upward to achieve regeneration and renewal of the epidermis, and their proliferation and differentiation are also essential for maintaining skin homeostasis during skin expansion. Usually, mechanical stimulation induces keratinocyte death and inflammation in the skin and is thought to be detrimental to keratinocytes [90,114]. However, during skin soft tissue expansion, mechanical stimulation is beneficial for promoting basal keratinocyte proliferation. This is found to be associated with calcium influx, EGFR, ERK1/2 cascades, and YAP signaling [74,114]. In vitro, Takei et al. found that the proliferation rate of human keratinocytes increased significantly under cyclic strain, but not under constant strain [115]. Further experiments by Kippenberger et al. found that mechanical stretch protects cells against apoptosis by inducing activation of the PKB/Akt pathway through transactivation of EGFR [116]. Therefore, the overall effect of proper mechanical stretching on keratinocytes is to increase cell division and decrease apoptosis, which is the basis of epidermal expansion. There are three main sources of keratinocytes in the basal layer: the interfollicular epidermis stem cells of the basal layer, the hair follicle bulge-derived stem cell, and the sebaceous gland (SG) stem cells located above the bulge and below the hair shaft orifice [117]. The interfollicular stem cells divide asymmetrically, giving rise to a subset of daughter cells that migrate upward as they differentiate, eventually losing their nuclei and forming cornocytes [118]. Recently, Kretzschmar et al. identified Troy as a new marker of interfollicular epidermis stem cell in developing and adult human and mouse epidermis via single-cell transcriptomics. This study also suggests that Troy-expressing basal cells contribute to the renewal of the epidermis [119].

### 4.2. Fibroblast in Dermis under Mechanical Stretch

Dermal fibroblasts are the main resident cells in the dermis, secreting collagen I and producing extracellular matrix to maintain skin structure and strength. Previous studies have shown that dermal fibroblasts are the key cells in wound healing, but also the main cause of scar formation [120]. Mechanical stimulation is an important factor in regulating the activity of dermal fibroblasts. Skin mechanics is related to scar formation [121]. Aarabi et al. applied mechanical stretch to mouse wounds, which resulted in the proliferation of fibroblasts, increased synthesis of extracellular matrix, and a hypertrophic scar appeared on the skin of mice [122]. Webb et al. studied the effect of cyclic strain (10% Strain, 0.25Hz, 8 h/day) on human fibroblasts in vitro and found that the proliferation of fibroblasts increased significantly, accompanied by an accumulation of the extracellular matrix [123]. Cyclic strain induces changes in the synthetic function of fibroblasts, mainly showing a significant increase in collagen I synthesis, changes in expression of remodeling enzymes (down-regulated MMP-1 and up-regulated TIMP-1), and increased expression of some growth factors (TGF-β, CTGF) [123,124]. Dermal fibroblasts are subjected to similar mechanical stimulation during skin soft tissue expansion. After the expander was implanted, fibroblasts were stimulated mechanically and began to proliferate, resulting in increased cell density in the dermis [125]. Fibroblast activity is also increased and more collagen is synthesized, mostly parallel to the surface of the expander [9]. Electron microscopy showed that there were more rough endoplasmic reticula and mitochondria in proliferating fibroblasts and some of them were swollen, which might promote cell proliferation [8].

### 4.3. Mesenchymal Stem Cells under Mechanical Stretch

Mesenchymal stem cells (MSCs) are multipotent cells with the ability to differentiate into a variety of cells, including chondroblasts, osteoblasts, and adipocytes [126]. They come from a wide range of sources, including bone marrow, adipose tissue, umbilical cord, placenta, and hair follicle. When skin is injured, both bone marrow mesenchymal stem cells (BMSCs) and adipose-tissue-derived stem cells (ADSCs) can be rapidly recruited to the injured site and differentiate into dermal fibroblasts, endothelial cells, and keratinocytes to participate in skin regeneration [127]. Due to the role of mesenchymal stem cells in tissue regeneration, previous works from our group and Li qingfeng’s group demonstrated that mesenchymal stem cells under mechanical stretch can promote skin growth and shorten the process of skin expansion, raising more attention to the mechanistic study of MSCs promoting skin regeneration during skin soft tissue expansion [128,129].

BMSCs are isolated from bone marrow and can be self-derived without immune rejection. Animal studies in murine animals and pigs have shown that mechanical stretch during skin soft tissue expansion promotes skin expansion by recruiting more BMSCs and facilitating their transdifferentiation [130]. A chimeric mouse model in which enhanced green fluorescent protein (EGFP) transgenic mouse bone marrow was transplanted into C57BL mice with the same genetic background through the tail vein was used in our previous work. We found that more BMSCs were recruited into the expanded skin and converted into vascular endothelial cells, epidermal cells, and spindle-shaped dermal fibroblasts [128]. In the pig experiment, it was found that local injection had a higher expansion effect than intravenous injection and BMSCs also promoted the secretion of growth factors, VEGF, bFGF, EGF, and SDF [131,132]. The way in which BMSCs promote expanded skin regeneration is illustrated in Figure 3.

ADSCs are mesenchymal stem cells found in the subcutaneous tissue and are one of the main sources of extracellular matrix proteins involved in maintaining skin structure and function [127]. Stromal vascular fraction (SVF) derived from ADSCs is widely used in cosmetic surgery [133]. Li et al. transplanted SVF into the expanded skin of Wistar rats and found that SVF significantly improved the expansion efficiency and up-regulated the expressions of EGF, VEGF, and bFGF [134]. Recently, a randomized, controlled clinical trial of autologous stromal vascular fraction cells transplantation was conducted [135]. In this study, twenty patients participated in the trial and the results showed that the expanded skin receiving SVF grafts was thicker and more proliferating cells, more blood vessels, and more extracellular matrix were observed. However, the clinical application of SVF is limited due to its traditional collagenase-based production. In addition, An et al. studied the effect of emulsified fat (EF), another source of ADSCs, on expanded skin during tissue expansion. The results showed that under the condition of consistent pressure in the expander (60 mmHg), the skin area of the rats treated with emulsified fat expanded more rapidly and the higher the treatment dose, the higher the skin expansion efficiency [136]. Consistent with wound healing, ADSCs promote skin expansion mainly through differentiation into fibroblasts, keratinocytes and endothelial cells, and producing growth factors [127,134]. The way in which ADSCs promote skin soft tissue expansion is illustrated in Figure 3.

Hair-follicle-bulge-derived stem cells (HFBSCs) can differentiate into keratinocytes, nerve cells, smooth muscle cells, etc., and are identified as mesenchymal stem cells [137,138]. HFBSCs are closely related to wound healing. Ito et al. found that HFBSCs migrate to trauma centers in a linear manner after skin damage and develop epidermal cell phenotypes [139]. Therefore, HFBSCs may participate in skin regeneration by migrating to epidermal sites and differentiating into epidermal cells [140]. In view of the high potential of HFBSCs to promote skin regeneration, the role of HFBSCs in promoting skin expansion induced by mechanical stretch was also validated. We previously found that the topically injected HFBSCs significantly improved the efficiency of skin expansion via differentiation into endothelial cells, epidermal cells, and the outer root sheath cells of hair follicles, up-regulating EGF, VEGF, bFGF, and TGF-β expression [47]. Wnt signaling plays a crucial role in regulating HFBSCs. Wnt signaling activation can facilitate re-epithelialization, hair follicle formation, and promote HFBSCs to differentiate into sebaceous gland cells [141,142]. The way in which HFBSCs promote skin soft tissue expansion is also illustrated in Figure 3.

### 4.4. Chemical and Mechanical Regulation of Mesenchymal Stem Cells

Mesenchymal stem cells need to migrate from their original niche to the site of tissue expansion to proliferate and differentiate to promote skin growth and regeneration. This complex process is influenced by many factors, including chemical factors (such as chemokines, cytokines, growth factors) and mechanical factors (such as mechanical stretch and extracellular matrix stiffness) [143].

Many studies have shown that SDF-1/CXCR4 axis is essential for inducing BMSC homing and participates in tissue regeneration [132]. Studies have found that SDF-1 gene expression is up-regulated in expanded skin after receiving BMSC grafts [128,131,132]. A recent study showed that SDF-1 promoted the migration, proliferation, and differentiation of BMSCs through the Wnt/β-catenin pathway [144]. The MMP-9/CXCR8 axis is another important factor for inducing the homing of BMSCs [145]. Transcriptome sequencing result showed that both MMP-9 and CXCR8 genes were up-regulated, suggesting their potential role in promoting BMSCs homing in mechanical-stretch-induced skin regeneration [81]. Growth factor is a kind of polypeptide functioning in regulating cell migration, proliferation, and differentiation. Increased expression of growth factors bFGF, VEGF, and EGF has been found in expanded skin after mesenchymal stem cell transplantation [131,134]. bFGF promotes the migration and differentiation of BMSCs by activating the Akt/Protein kinase B (PKB) pathway at low concentrations but inhibits BMSCs at high concentrations [146]. VEGF, usually referred to as VEGF-A, is a specific mitogen of vascular endothelial cells, which promotes angiogenesis and increases vascular permeability. VEGF promotes skin soft tissue expansion in two ways. Firstly, VEGF promotes recruitment of BMSCs by stimulating platelet-derived growth factor receptors (PDGFRs) and increasing SDF-1 level [147,148]. Secondly, VEGF promotes angiogenesis by promoting the differentiation of BMSCs into vascular endothelial cells, which can improve the blood and nutrient supply in expanded skin [149]. Further studies showed that the upregulation of VEGF in expanded skin was related to the Janus kinase/signal transducer and activator of transcription (Jak-STAT) and Wnt signaling pathway [130]. EGF is an important growth factor that enhances cell proliferation activity. EGF not only induces MSC proliferation through EGFR/ERK and AKT pathways, but also promotes MSC proliferation, invasion, and migration by inducing expression of MSC-derived exosomes microRNA-21 (miR-21) [150,151].

In vitro experiments showed that cyclic stretching stimulation can promote the proliferation, adhesion, and migration of human ADSCs, regulate their differentiation and inhibit apoptosis [152]. In vivo, mechanical stretch can promote skin tissue regeneration via enhancing MSC homing and transdifferentiation and modulates genes related to skin hypoxia, vascularization, and cell proliferation [130]. However, through cell experiments, Zhang et al. found that BMSC invasion was inhibited, despite increased BMSC migration under cyclic mechanical stretch [153]. They found that the FAK-ERK1/2 signaling pathway activated by mechanical stretch is critical for promoting BMSC migration. At the same time, FAK reduced BMSC invasion by reducing matrix metalloproteinase 2 (MMP-2) and matrix metalloproteinase 9 (MMP-9). Matrix metalloproteinases (MMPs) can decompose the extracellular matrix and play an important role in cell invasiveness. Using short-interfering RNA to interfere in MMP gene expression can reduce the homing and invasion of MSC [130]. Recent studies have shown that the PI3K/Akt signaling pathway is also involved in the invasion of MSCs. Mechanical stretch inhibited the PI3K/Akt signaling pathway and, thus, down-regulated the expression of membrane type-1 matrix metalloproteinases (MT1-MMP) at both the mRNA and protein levels [154].

### 4.5. Clinical Application of Mesenchymal Stem Cells

There are hundreds of clinical trials using mesenchymal stem cells and their derivatives to treat human diseases, showing great therapeutic potential. Bone marrow mesenchymal stem cells (BMSCs) are the most representative of them, accounting for nearly half of all clinical trials [155]. At the same time, there has been an increasing interest in adipose mesenchymal stem cells for skin therapy [127]. Therefore, we cannot help but expect their great roles in promoting mechanical-stretch-induced skin regeneration during skin soft tissue expansion.

Both BMSC- and ADSC-induced skin growth and regeneration have been clinically tested and shown satisfactory results in skin expansion [135,156]. After BMSC and ADSC transplantation in patients, the expanded skin showed larger area, thicker epidermis and dermis, more blood vessels, and higher expansion index (EI). EI, the total volume inflated divided by the designed volume of expander, was defined by Zhou et al., which is used to evaluate the expansion degree of expander with different specifications [156]. For the patients subjected to skin soft tissue expansion preparing for defect repair and organ reconstruction, isolating ADSCs from self-extracted fat to promote expanded skin regeneration would be acceptable. In addition, for many beauty seekers, excess fat affects their body shape and they are happy to extract their own fat for medical purposes. However, Krasilnikova et al. showed that mesenchymal stem cells alone have limited skin regeneration ability, so we just recommend it to be a potential adjuvant therapy to promote skin soft tissue expansion [157]. Recent studies have shown that adipose-cell-free derivatives are the main factor at play, including ADSC-conditioned medium (ADSC-CM), ADSC exosomes (ADSC-Exo), and cell-free adipose tissue extracts (ATEs) [158,159,160]. They have lower immunogenicity and are easier to store and transport. Therefore, adipose-cell-free derivatives may be a promising means to improve the efficiency of skin expansion in the future.

### 4.6. Cellular Strategies to Improve Skin Soft Tissue Expansion

Normally, skin-derived stem cell differentiation and proliferation of keratinocytes and fibroblasts contribute to expanded skin regeneration under mechanical stretch. However, their ability to differentiate and proliferate is limited, as indicated by the slow rate of expansion and the thinning of the epidermis. Therefore, our aim should concentrate on enhancing the differentiation and proliferation of skin cells. There are many ways to promote cell differentiation and proliferation, but few have been tested in skin soft tissue expansion, which should be done later. Moreover, increasing studies have found that MSCs are important sources of keratinocytes and fibroblasts and may serve as a supplementary seed cell source during skin regeneration. Although the contribution of several types of MSCs to skin regeneration has been studied, the roles and functions of many other MSCs in skin regeneration remain unclear. Hence, more studies should be conducted to investigate the cellular sources of skin regeneration, thus, improving skin soft tissue expansion.

With the advent and evolution of CRISPR and other gene-editing platforms, stem cells based on gene-editing technologies may also become candidate treatment strategies in the future and hold great promise for improving skin soft tissue expansion. On the one hand, gene-editing technologies can reduce the immunogenicity of stem cells for allogeneic use, making non-invasive harvesting possible. On the other hand, stem cells based on gene-editing technologies can also act as therapeutic agents to promote skin soft tissue expansion by overexpressing various kinds of growth factors and other molecules that assist in tissue recovery [161]. However, we also should pay great attention to the biosafety of exogenous MSCs and gene-editing technologies; their potential tumorigenic and off-target effects are detrimental to patients.

## 5. Methods to Promote Skin Regeneration under Mechanical Stretch during Skin Soft Tissue Expansion

At present, several methods have been used to promote skin growth during skin soft tissue expansion. Firstly, different expansion procedures have different stimulative effects due to the difference in intensity and duration of mechanical stretch applied to the skin. Hence, selecting appropriate procedures may benefit tissue expansion. In addition, local injection of Botulinum toxin TYPE A (BTXA) can increase skin expansion by inhibiting fibrous capsule formation and increasing blood flow [162]. Additionally, multiple growth factors have also been found to be effective in promoting skin growth, such as EGF and VEGF [134]. Containing many important growth factors, platelet-rich plasma (PRP) is also an effective growth-promoting drug [163]. Compared with these methods, here, we emphasize BMSC and ADSC transplantation, which can not only secrete a variety of growth factors, but also proliferate and differentiate to provide additional cells for skin to achieve double-promoting effects.

Based on our limited understanding, we speculate that focusing on the following methods may facilitate expanded skin regeneration: (1) regulating signaling pathways and ion channels to modulate cell behaviors, (2) transplantation of exogenous MSCs or activating local in vivo MSCs, and (3) enhancing cell activity using exogenous growth factors. These new approaches should be based on the key molecules, critical signaling pathways, and promising mesenchymal stem cells that have been extensively studied.

## 6. Conclusions

In conclusion, in order to achieve high-quality skin in a shorter time, ultimately, regulating the activity of related signaling pathways and/or the behavior and activity of skin cells during skin soft tissue expansion may serve as a candidate choice. All of these are based on in-depth studies on how mechanical stretch promoting expanded skin regeneration during tissue expansion. Further study should focus on revealing the source of cells that contributes to skin regeneration and how mechanical stretch regulates various intracellular signaling pathways in expanded skin. Meanwhile, the clinical application potential of mesenchymal stem cells, cell-free derivative products, and growth factors should be further investigated.

## Figures and Tables

**Figure 1 ijms-23-09622-f001:**
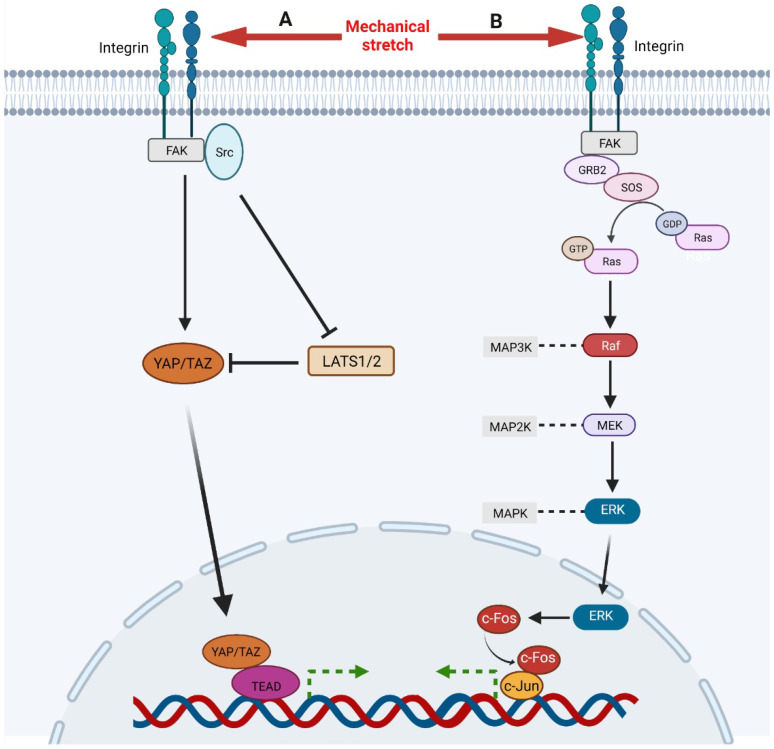
A, Mechanical stimulation, integrin-recruited FAK, and Src activate YAP/TAZ directly and facilitate its nuclear translocation. Meanwhile, FAK and Src can also inhibit LATS1/2, an inhibitor of YAP/TAZ, to activate YAP/TAZ signaling. Then, activated YAP/TAZ translocates to nuclear and binds to TEAD to regulate gene expression. B, Mechanical stimulation activates integrin which recruits and activates FAK. Then FAK activates Ras through GRB2 and SOS, initiating Ras-Raf-MEK-ERK cascade. Ultimately, activated ERK translocates to nuclear to up-regulates and activates c-Fos which binds c-Jun to regulate gene expression.

**Figure 2 ijms-23-09622-f002:**
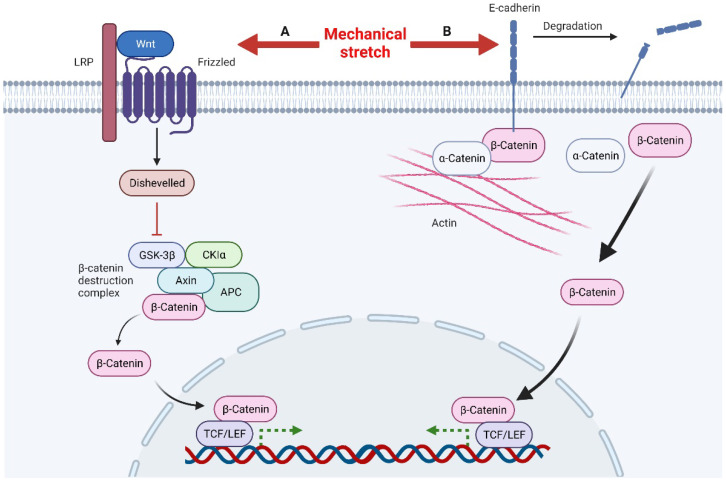
Mechanical stretch induces β-catenin activity through Wnt-protein-dependent and -independent ways. A, Mechanical stretch up-regulates Wnt proteins to activate Wnt signals. After Disheveled is recruited to inactivate the β-catenin destruction complex, this leads to the accumulation of β-catenin in the nucleus. β-catenin combines with the transcription factor TCF/LEF to regulate the transcription of target genes. B, Mechanical stimulation can also lead to the degradation of E-cadherin, releasing β-catenin in the E-cadherin binding pool, resulting in the accumulation of β-catenin in the nucleus and ultimately regulating the transcription of target genes.

**Figure 3 ijms-23-09622-f003:**
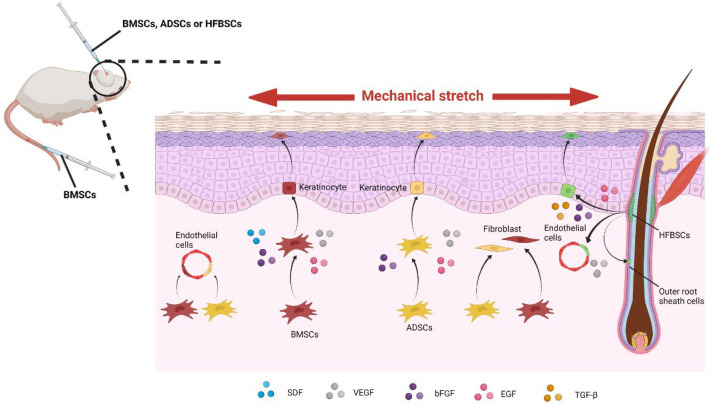
Under mechanical stretch, the injected bone marrow mesenchymal stem cells (BMSCs) and adipose-tissue-derived stem cells (ADSCs) differentiate into keratinocytes, fibroblasts, and endothelial cells and secrete various kinds of growth factors. In addition, hair-follicle-bulge-derived stem cells (HFBSCs) also differentiate into keratinocytes, endothelial cells, and the outer root sheath cells and secrete growth factors under mechanical stretch.

## Data Availability

Not applicable.

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
