# Peer review of "Mechanical Stretch Induced Skin Regeneration: Molecular and Cellular Mechanism in Skin Soft Tissue Expansion"

_ijms, 2022, doi:10.3390/ijms23179622_

Round 1

Reviewer 1 Report

1. The presented paper reviews the molecular and cellular mechanisms of skin regeneration induced by mechanical stretch during skin soft tissue expansion. Mechanical stimulation and mechanotransduction are the promised techniques for external cell regulation approach. The authors carefully discuss molecular and cellular mechanisms, and concluded that it needs methods to promote skin regeneration under mechanical stretch. 2. The manuscript' sections should be more detailed and contain the latest insights into the mechanobiology of the skin regeneration. I recommend to note some recently related papers with novel findings and ideas that may improve this paper: 2.1. (1. Introduction) "The dynamic fluctuations in force and shape that happen in vivo by actively stimulating cell–material constructs with controlled mechanical deformations" [Guimarães, C. F., Gasperini, L., Marques, A. P., & Reis, R. L. (2020). The stiffness of living tissues and its implications for tissue engineering. Nature Reviews Materials, 5(5), 351-370.] 2.2. (3.2. Fibroblast in dermis under mechanical stretch) "Skin mechanics is related to the problem of scar formation" [Yannas, I. V., & Tzeranis, D. S. (2021). Mammals fail to regenerate organs when wound contraction drives scar formation. NPJ Regenerative Medicine, 6(1), 1-6.] 2.3. (3.5. Clinical implication of mesenchymal stem cells) "Mesenchymal stem cells therapy alone does not lead to complete restoration of damaged skin, and it's essential to use adjuvat stimulation by biomaterials or biomolecules" [ Maksimova N.V., Michenko A.V., Krasilnikova O.A., Klabukov I.D., Gadaev I.Y., Krasheninnikov M.E., ... & Lyundup A.V. (2022). Mesenchymal stromal cell therapy alone does not lead to complete restoration of skin parameters in diabetic foot patients within a 3-year follow-up period. BioImpacts, 12(1): 51-55. https://doi.org/10.34172/bi.2021.22167  ] 2.4. (4. Methods to promote skin regeneration under mechanical stretch during skin soft tissue expansion) “Stem cell injection leads to wound healing primarily through stimulation of angiogenesis” [Krasilnikova O.A., Baranovskii D.S., Lyundup A.V., Shegay P.V., Kaprin A.D., Klabukov I.D. (2022). Stem and Somatic Cell Monotherapy for the Treatment of Diabetic Foot Ulcers: Review of Clinical Studies and Mechanisms of Action. Stem Cell Reviews and Reports, 1-12. https://doi.org/10.1007/s12015-022-10379-z ] 2.5. The listed papers with advanced approaches for skin regeneration and tissue engineering should be discussed in the paper (4. Methods to promote skin regeneration under mechanical stretch during skin soft tissue expansion): - Gómez-Gálvez, P., Anbari, S., Escudero, L. M., & Buceta, J. (2021, December). Mechanics and self-organization in tissue development. In Seminars in Cell & Developmental Biology (Vol. 120, pp. 147-159). Academic Press. - Waters, S. L., Schumacher, L. J., & El Haj, A. J. (2021). Regenerative medicine meets mathematical modelling: developing symbiotic relationships. NPJ Regenerative Medicine, 6(1), 1-8. - Boschetti, F. (2022). Tissue Mechanics and Tissue Engineering. Applied Sciences, 12(13), 6664. - Zha, S., Utomo, Y. K. S., Yang, L., Liang, G., & Liu, W. (2022). Mechanic-Driven Biodegradable Polyglycolic Acid/Silk Fibroin Nanofibrous Scaffolds Containing Deferoxamine Accelerate Diabetic Wound Healing. Pharmaceutics, 14(3), 601. - Masri, S., & Fauzi, M. B. (2021). Current insight of printability quality improvement strategies in natural-based bioinks for skin regeneration and wound healing. Polymers, 13(7), 1011. - Kimura, S., & Tsuji, T. (2021). Mechanical and immunological regulation in wound healing and skin reconstruction. International Journal of Molecular Sciences, 22(11), 5474.   -  Kozaniti, F. K., Deligianni, D. D., Georgiou, M. D., & Portan, D. V. (2021). The role of substrate topography and stiffness on MSC cells functions: Key material properties for biomimetic bone tissue engineering. Biomimetics, 7(1), 7. -  Aleemardani, M., Trikić, M. Z., Green, N. H., & Claeyssens, F. (2021). The importance of mimicking dermal-epidermal junction for skin tissue engineering: a review. Bioengineering, 8(11), 148. - Zemel, A. (2015). Active mechanical coupling between the nucleus, cytoskeleton and the extracellular matrix, and the implications for perinuclear actomyosin organization. Soft matter, 11(12), 2353-2363.

Reviewer 2 Report

The paper is a good review of what we know about tissue expansion and the bioprocesses that are affected by mechanical stress as well as the ones that can affect skin regeneration. The goal however, appears to be how to design a “better” treatment. A big issue is that the studies cited are hypothesis driven and only need to show that something is statistically better or there is a correlation between bioprocesses. To design a better treatment it has to meet the desired clinical outcome that is currently not being met. If the desired outcome is already met “better” is meaningless as well as if the better treatment still does not meet the desired outcome “better” is also meaningless.   

Therefore, a more design approach (engineering design process) is needed for the review, to be helpful in designing a “better” treatment. The information in the review is probably comprehensive enough, but it just needs to be in a more design organized framework. This approach will also show what information we do not currently know vs. the strategy laid out in section 4 about the potential of adding stem cells and growth factors in the hopes of getting a “better” response.

The first step in the engineering design process is identifying the problem. This can be taken from the Introduction and the Conclusion and appears to be that both the speed of the process and the quality of the expanded tissue are not good enough. Other things mentioned were high retraction rate as well as other complications. From this, we can come up with the need statement (what current tissue expanders don’t do) and what success would look like (Clinical Performance Design Constraints). Again, you don’t pick a treatment because it is “better”, it has to meet the desired clinical outcomes (including ones that are not currently being met). The Clinical Performance Design Constraints should be both a minimum required benefit (speed and quality) and a maximum acceptable risk (complications--including retraction). There also needs to be a design hierarchy with the top being the Clinical Performance design constraints then the system performance design constraints (what the treatment needs to do to meet the clinical performance design constraints) with each level down, what is required to meet the level above. In this review, it could be organized as what response is needed at the molecular level to get the desired cellular response to get the desired tissue response to get the desired clinical outcome.

Therefore, the structure of the review could be similar with the Introduction setting the stage for the purpose of the paper and the organization; explaining the design process (what the problem is and what success looks like at the clinical level). Then the desired bioprocesses should be selected at each level (macro—tissue, then cells, then molecular level). The review will talk about the bioprocesses starting from the bottom and then discuss strategies to get the desired response at each level; including studies that are needed to do to determine the appropriate bioprocesses at each level.

There then should be a new section, which would build on the Introduction and give the specific problem and where we are at currently followed by what success would look like (the clinical performance design constraints) as well as the desired tissue response and where we are currently (which was covered in what normally happens to tissue after expansion). The next two sections (molecular and cellular levels) would be similar to what they are now but organized slightly different. For the molecular pathways, present what we know about them and how they are involved in tissue expansion and tissue regeneration. Then how they could be altered to get a more desirable cellular and tissue response as well as potential strategies to do this (or what we need to do to find this out). Next is what cells normally do under stretch and how it is related to the molecular pathways (which was done) plus add thoughts about desired responses (building on the previous section) and how to achieve them (or what we need to do to find this out). The stem cell strategies should go in the last part of this section.

Section 4 (which is now section 5) should be what we know about improvements at the cell and molecular level to get the desired macro response to get the desired clinical response (or what we need to know). It should be only a minimal overlap with the previous sections, but decide how much of this goes in the previous sections and how much here. It could be more what in general to do to modify the molecular and cellular responses in the previous sections and the specific strategies here (which would include the stem cell and growth factor approaches currently in section 4).

The conclusion would summarize what is needed to reach the clinical performance requirements, which would include both strategies and additional studies to determine the desired bioprocesses at each level vs. what there is currently in the Conclusion:

“In conclusion, the mechanisms of skin soft tissue expansion are far from being explored, especially for those promote skin growth and regeneration. Our ultimate goal is to achieve high quality skin in a shorter time and bettering the treatment experience of patients. Further study should focus on revealing the source of cells that contributes to skin regeneration and how mechanical stretch regulates various intracellular signaling pathways in expanded skin. Meanwhile, the clinical application potential of mesenchymal stem cells and cell-free derivative products should be further developed.”

For example, the specific types of studies and information that is needed to complete the design hierarchy and suggest potential strategies can be presented (vs. just general areas). This will include a having a better understanding of the desired processes at the molecular level to give the desired bioprocesses at the cellular level to give the desired clinical performance constraints. It appears that currently only the relationships and correlations are known between bioprocesses at different levels as well as the potential relationships between strategies and changes in bioprocesses. Therefore, there are probably a number of studies needed to turn the relationships into quantifiable changes.

Reviewer 3 Report

The work by Guo and collaborators is a very well written Review of the literature dealing with the molecular and cellular bases of skin regeneration stimulated by mechanical stretch during skin soft tissue expansion surgery.

The manuscript is very well systematized, well written and easy to read. The figures are very clear and aid in understanding the different intracellular/molecular signalling pathways which are associated to mechanical stretch stimulation of skin regeneration.

The work is of high quality and deserves to be accepted in the present form for publication.

Author Response

Dear reviewer:

We gratefully thank you for the precious time you spent making constructive remarks. Your deep and kind comments have helped us a lot. And we are very glad to know your acceptance of our manuscript.

Thanks again for your insightful comments.

Sincerely yours,

Zhou Yu

Reviewer 4 Report

1. Section "Intracellular signaling under mechanical stretch", the crosstalking network of signaling pathways need to be provided.

2. In addition to cellular adhesion molecules, growth factors and stretch-activated ion channels are also capable of sensing mechanical stimuli. Please provide relevant research progress.

Reviewer 5 Report

Skin soft tissue expansion is one of the most commonly used techniques in medical practice. This article reviews the molecular and cellular mechanisms of skin regeneration induced by mechanical stretch during skin soft tissue expansion, which facilitate better understanding of the mechanisms and may provide new sight of this procedure. However, this manuscript would be better presented if clear rationale were included.
Specific Comments: 
1. Some of the expressions within the manuscript are unclear and poor to understand, such as “feel for a variety of uses” in line 27; “excess fat is a drag” in line 545, should be elaborated and corrected.

2. As described in 2.1.3 and 3.1, unlike in vitro experiment and in vivo animal experiment with cyclic strain, mechanical stretch during skin soft tissue expansion in actual clinal situation are sustainable and long-lasting, usually lasts several months, may have more profound effect on molecular and cellular of the skin and soft tissue. How do you recognize this difference?

3. In line 464-479, the author described and discussed how hair follicle bulge-derived stem cells (HFBSCs) participant in skin regeneration, however, most of the cited literature mainly concentrate on the motivation and activation of stem cells after epidermal injury, which may be different from the situation of skin soft tissue expansion. More convincing evidence should be presented from latest articles. Besides, under physiological conditions, human hair follicle dermal papilla cells (HFDPC) are the main cells sources responsible for the production of hair growth and the development of hair follicles, hair follicle bulge-derived stem cells (HFBSCs) would participant in epidermis in specific conditions (results from Ito et al). Hence, skin soft tissue expansion induced hair follicle regeneration should be reconsidered, further studies are required.

4. Stem cell therapy has been proven a safe, effective and promising method for several decades, the harvesting of the BMSCs and ADSCs for transplantation could be invasive and result in lower acceptance. Stem cells based on gene-editing technology, or noninvasive harvesting could also be concerned. 

Round 2

Reviewer 4 Report

Accept in present form